# The Enigma of Mammaglobin: Redefining the Biomarker Paradigm in Breast Carcinoma

**DOI:** 10.3390/ijms241713407

**Published:** 2023-08-29

**Authors:** Bojan Milosevic, Bojan Stojanovic, Aleksandar Cvetkovic, Ivan Jovanovic, Marko Spasic, Milica Dimitrijevic Stojanovic, Vesna Stankovic, Marija Sekulic, Bojana S. Stojanovic, Natasa Zdravkovic, Minja Mitrovic, Jasmina Stojanovic, Darko Laketic, Maja Vulovic, Danijela Cvetkovic

**Affiliations:** 1Department of Surgery, Faculty of Medical Sciences, University of Kragujevac, 34000 Kragujevac, Serbia; drbojanzm@gmail.com (B.M.); bojan.stojanovic01@gmail.com (B.S.); draleksandarcvetkovic@gmail.com (A.C.); 2Center for Molecular Medicine and Stem Cell Research, Faculty of Medical Sciences, University of Kragujevac, 34000 Kragujevac, Serbia; ivanjovanovic77@gmail.com; 3Department of Pathology, Faculty of Medical Sciences, University of Kragujevac, 34000 Kragujevac, Serbia; milicadimitrijevic@yahoo.com (M.D.S.); wesna.stankovic@gmail.com (V.S.); 4Department of Hygiene and Ecology, Faculty of Medical Sciences, University of Kragujevac, 34000 Kragujevac, Serbia; msekulic82@gmail.com; 5Department of Pathophysiology, Faculty of Medical Sciences, University of Kragujevac, 34000 Kragujevac, Serbia; 6Department of Internal Medicine, Faculty of Medical Sciences, University of Kragujevac, 34000 Kragujevac, Serbia; natasasilvester@gmail.com; 7Department of Neurology, Faculty of Medical Sciences, University of Kragujevac, 34000 Kragujevac, Serbia; minjam034@gmail.com; 8Department of Otorhinolaryngology, Faculty of Medical Sciences, University of Kragujevac, 34000 Kragujevac, Serbia; fonijatarkg@gmail.com; 9Institute of Anatomy, Faculty of Medicine, University of Belgrade,11000 Belgrade, Serbia; drdarkolaketic@gmail.com; 10Department of Anatomy, Faculty of Medical Sciences, University of Kragujevac, 34000 Kragujevac, Serbia; maja@medf.kg.ac.rs; 11Department of Genetics, Faculty of Medical Sciences, University of Kragujevac, 34000 Kragujevac, Serbia; c_danijela@yahoo.com

**Keywords:** mammaglobin, breast carcinoma, biomarker, tumoral expression, peritumoral expression, cancer progression, metastasis, diagnosis, treatment strategies

## Abstract

The continuous evolution of cancer biology has led to the discovery of mammaglobin, a potential novel biomarker for breast carcinoma. This review aims to unravel the enigmatic aspects of mammaglobin and elucidate its potential role in redefining the paradigm of breast carcinoma biomarkers. We will thoroughly examine its expression in tumoral and peritumoral tissues and its circulating levels in the blood, thereby providing insights into its possible function in cancer progression and metastasis. Furthermore, the potential application of mammaglobin as a non-invasive diagnostic tool and a target for personalized treatment strategies will be discussed. Given the increasing incidence of breast carcinoma worldwide, the exploration of novel biomarkers such as mammaglobin is crucial in advancing our diagnostic capabilities and treatment modalities, ultimately contributing to improved patient outcomes.

## 1. Introduction

Breast carcinoma is one of the most prevalent malignancies affecting women worldwide, posing significant challenges to early detection, accurate prognosis, and effective treatment [1,2]. Despite considerable advances in medical technologies, the quest for reliable and sensitive biomarkers that can aid in the early detection and monitoring of breast carcinoma remains of paramount importance [3]. Biomarkers not only provide insights into the pathophysiology of diseases but can also guide individualized treatment strategies, hence contributing to the overall improvement of patient outcomes [4].

In this intriguing quest, mammaglobin, a member of the secretoglobin family, has emerged as a potential novel biomarker [5]. Mammaglobin, also known as mammaglobin-A or SCGB2A2, belongs to a cluster of small secretory proteins [6]. It was first discovered in 1996 through a differential screening of a human breast carcinoma cDNA library, with its expression being predominantly limited to the mammary gland [7,8]. While its exact function remains somewhat elusive, mammaglobin is believed to play roles in various biological processes, including cell signaling, immune response, chemotaxis, and potentially hormone transport [8]. The protein is known to bind steroid-like molecules and is part of the secretoglobin family, which comprises over 20 related proteins [6,8]. The *SCGB2A2* gene, located on chromosome 11q13, encodes mammaglobin, a 93-amino-acid secretory protein [5]. It is believed to play a role in the secretory processes of the mammary gland under normal circumstances [8]. However, it is frequently upregulated in instances of breast cancer, emphasizing its potential as a therapeutic target for these tumors [9].

Human mammaglobin (hMAG) is primarily a small epithelial secretory protein and a member of the uteroglobin/Clara cell protein family [6,10]. It possesses two N-linked glycosylation sites, contributing to the addition of approximately 3 kDa carbohydrate chains [11]. In breast tissue, hMAG-A protein exists in two forms with molecular masses of around 18 kDa and 25 kDa, both of which are more commonly found in breast carcinomas compared with fibroadenomas or normal breast tissues [9,11]. Interestingly, the expression of mammaglobin appears to be independent of steroid hormones and is potentially regulated by mechanisms involving Phosphoprotein Enriched in Astrocytes 3 (PEA-3) expression [9]. Molecular modeling studies suggest a structural arrangement of mammaglobin, with four alpha helices per protein forming a hydrophobic core in a head-to-tail orientation [9]. The presence of N-linked glycosylation sites at opposite ends of the complex, as well as in flexible loop regions between the alpha helices, allows for the extension of large carbohydrate chains in either direction [12]. Additionally, mammaglobin forms covalent heterodimers with lipophilin B, exhibiting an anti-parallel arrangement that facilitates the formation of three disulfide bridges between them [13]. 

Mammaglobin overexpression, particularly specific to breast cancer cells, lends its compelling potential as a unique biomarker for breast carcinoma [14]. Its detectability in peritumoral tissue and circulation suggests possible uses in a variety of clinical applications, including diagnosis, prognosis, and potential therapeutic targeting [9,15]. Yet, the mechanisms driving its overexpression and its role in tumorigenesis and cancer progression are subjects of ongoing research. Although variably detected on the membrane fraction of breast cancer cells and present in both normal and cancer cell surfaces and cytoplasm, the precise function and significance of mammaglobin-A in breast cancer pathogenesis are yet to be fully understood [12].

Despite these gaps, mammaglobin-A’s clinical relevance is undeniable given its overexpression in breast cancer [7]. Its potential utility as a diagnostic and prognostic biomarker is backed by numerous studies linking mammaglobin levels with clinical parameters like tumor stage and lymph node status [16,17]. 

This review seeks to demystify mammaglobin, a potential gamechanger in the landscape of breast carcinoma biomarkers. It explores mammaglobin’s expression in tumoral and peritumoral tissues and its circulating levels in blood, along with its implications in cancer progression and metastasis. As a prospective non-invasive diagnostic tool, mammaglobin could revolutionize early cancer detection and prognostic prediction and guide therapeutic strategies [3]. Given the global rise in breast carcinoma, examining innovative biomarkers like mammaglobin is crucial.

## 2. The Pivotal Role of Biomarkers in Breast Carcinoma

Breast carcinoma is a significant global health concern, posing challenges in early detection, prognostication, and targeted therapies [18]. The risk of distant recurrence persists even five years post-diagnosis, necessitating the exploration of biomarkers for improved patient outcomes [19]. Biomarkers, measurable indicators of physiological or pathological processes, hold promise in revolutionizing breast cancer care by elucidating underlying disease mechanisms, facilitating early diagnosis, and tracking disease progression [20]. Their detection in non-invasive samples such as blood and urine enables early disease identification, thus promoting prompt intervention and a more favorable prognosis [3].

Various biomarkers like estrogen and progesterone receptors (ER and PR), gross cystic disease fluid protein-15 (GCDFP-15), and others have traditionally been employed in diagnosing and prognosticating breast cancer [21,22]. Yet, current markers, including carcinoembryonic antigen (CEA) and CA 15-3 (CA27-29), display limited sensitivity and specificity, necessitating improved diagnostic precision [23,24]. Numerous proposed biomarkers aim to improve detection but lack sufficient sensitivity and specificity to be clinically beneficial [3]. Prognostic indicators such as ER, PR, and HER2/neu and multigene panels like Oncotype DX have been instrumental in treatment planning and predicting disease outcomes, but they have limitations in accuracy and tissue specificity [25,26,27,28]. A comparative analysis of these traditional markers, including their specific remarks, can be found in Table 1. Consequently, research focuses on identifying innovative biomarkers with better sensitivity, specificity, and tissue specificity [29]. Human mammaglobin (hMAG), a promising candidate, is currently under investigation for its potential utility in diagnosis, prognosis, and therapy in breast cancer [9].

## 3. Expression Patterns of Mammaglobin-A in Diverse Human Tissues

Studies by Watson and Fleming explored mammaglobin-A mRNA expression in a variety of human tissues, both fetal and adult, primarily through RT-PCR and Northern blotting techniques [12,14]. They found that mammaglobin-A’s expression is predominantly restricted to the adult mammary gland, believed to be associated with the gland’s proliferation and terminal differentiation [12]. Further research corroborated the overexpression of mammaglobin-A in breast tumors, with about 80% exhibiting this characteristic [9]. This overexpression was found to be unrelated to breast carcinogenesis, but increased expression in breast tumors was associated with less aggressive tumor phenotypes and significantly higher in estrogen receptor-positive tumors [46].

Beyond breast tissue, mammaglobin-A has been detected in normal and cancerous tissues of the female genital tract and the sweat and salivary glands [8]. Its expression in breast tissue, however, significantly outpaces that in ovarian and endometrial tissues [8]. Immunostaining has revealed mammaglobin-A expression in a few normal tissue types like the luminal cells of the breast, endocervical glands, endometrium, and more [8,47]. Despite its presence in other tissues, mammaglobin-A’s expression is exceptionally specific to breast cancer, leading to its proposition as a potential marker for identifying circulating and disseminated tumor cells and verifying the breast origin of metastatic cancer [5,11]. 

## 4. Tumoral Expression of Mammaglobin

Mammaglobin, a potential biomarker for breast carcinoma, has drawn significant interest because of its overexpression in primary and metastatic breast cancer tissues and specificity to mammary tissue [7,9]. Numerous studies show high levels of mammaglobin in a considerable percentage of breast carcinomas, with variations based on tumor subtype and stage [8,46]. Its expression in other malignancies is largely minimal, heightening its appeal as a specific breast carcinoma biomarker [8]. There is a demonstrated correlation between high mammaglobin expression and adverse prognostic factors, implying a role in disease pathogenesis and progression [8,15]. However, the exact role and mechanisms of mammaglobin’s overexpression in breast carcinoma are unclear, and its non-universal overexpression could limit its standalone biomarker utility.

### 4.1. Mammaglobin in Breast Carcinoma Tissue: Its Potential Role as a Marker

Mammaglobin, known to exist in two isotypes, mammaglobin-A (MAM-A) and mammaglobin-B (MAM-B), presents distinct roles in cancer biology [48]. MAM-A, predominantly expressed in breast tissue, has been found in approximately 80% of breast tumors, showing an overexpression of up to 10 times compared with normal breast tissue [11,46]. This significant upregulation suggests that MAM-A is a potential molecular diagnostic marker for breast cancer [46]. Variability exists in the prevalence of MAM-A expression, with positivity rates ranging from 59% to 100% for lobular breast carcinomas and 25% to 94% for invasive breast carcinomas of no specific type [8]. On the other hand, MAM-B has been observed in a variety of cancer types, suggesting its role extends beyond breast cancer [46]. 

MAM-A, a highly specific marker for breast tissue, demonstrates significant potential, though it is not a perfect diagnostic marker, as it is not expressed in all breast cancer cell lines and tumors [5]. mRNA expression in mammaglobin in breast cancer cells is significantly elevated compared with non-malignant breast tissue [49]. This overexpression, likely modulated by intricate transcription mechanisms, has important implications [12]. Mammaglobin-A has been identified as an important predictor for bone metastases in breast cancer, and its expression pattern could aid in personalizing postoperative adjuvant treatment planning [15].

Interestingly, the mammaglobin protein complex in breast tumors appears to provoke an immune response, activating mammaglobin-reactive CD8^+^ and CD4^+^ T cells [50]. This immune response could reflect antitumor activity, with a potential correlation between the number of these cells and disease outcomes or recurrence. Antibodies against the mammaglobin complex have been found in the sera of breast cancer patients [11,51]. Levels of these antibodies correlate with disease stages, with higher antibody levels against a component called lipophilin B seen in advanced stages [52]. Antibodies specifically targeting mammaglobin are lower, possibly because of its high degree of glycosylation [11].

The overexpression of MAM-A is associated with high tumor grades, indicating its potential as a poor prognosis marker [53]. Numerous studies have demonstrated a positive correlation between MAM-A positivity and breast cancer stages [11,53]. Patients with MAM-A-positive invasive breast cancer typically present with larger tumors, higher histological grading, and a higher Ki-67 proliferation index, indicative of high cell proliferation [53].

The reduced expression of proteins typically found in tumor cells suggests tumor cell dedifferentiation, often linked to unfavorable tumor characteristics [54]. This might explain the associations between decreased MAM-A expression and high tumor grades or unfavorable molecular parameters in breast cancer [8]. Breast cancer patients with moderate MAM-A staining on their tumors have demonstrated the best patient outcomes, hinting that both the upregulation and downregulation of MAM-A could be linked to tumor progression [8]. The downregulation of MAM-A has been observed in 49% of breast cancers [8]. 

#### Mammaglobin Expression and Its Correlation with Hormone Receptor Status in Breast Cancer

Research has evidenced an association between elevated mammaglobin expression and the presence of pivotal hormone-responsive markers, specifically estrogen and progesterone receptors, in cases of breast cancer [51]. This elevated expression has also been connected to diploid DNA content, low cell proliferation rates, low nuclear grades, and the absence of axillary nodal invasion, hinting at a less aggressive tumor phenotype [11]. Additionally, several studies have indicated higher mammaglobin positivity rates in patients with estrogen receptor (ER)-positive tumors, suggesting a link between mammaglobin expression and estrogen responsiveness [53].

However, the connection between mammaglobin expression and ER status is complex, with some studies associating mammaglobin expression with negative ER status [55]. This inconsistency could be attributed to variations in study design, patient populations, or mammaglobin measurement methods. Consequently, further research is needed to elucidate the intricate relationship between mammaglobin and hormone receptor status in breast cancer.

### 4.2. Peritumoral Expression of Mammaglobin: A New Dimension in Breast Carcinoma Research

Peritumoral tissues, or tumor microenvironments, which are composed of various cell types and extracellular matrix components, play a vital role in cancer progression, influencing tumor growth, invasiveness, and metastatic potential [56]. The expression of mammaglobin in these tissues introduces an intriguing aspect to its potential as a biomarker [15]. Initial research has found mammaglobin to be present in these areas, suggesting a possible link to local invasion, metastasis, and aggressive disease phenotypes [15].

However, the exact role of mammaglobin in the peritumoral environment is still undefined. It is uncertain whether mammaglobin’s expression in these tissues results from the tumor’s influence or if it actively shapes the tumor microenvironment [15]. Likewise, the cell types expressing mammaglobin in the peritumoral environment and its potential role in modulating the local environment to support tumor growth and spread remain unclear. These questions highlight the need for further research in this area.

In a study investigating mammaglobin levels in breast cancer patients, it was found that mammaglobin concentrations were elevated in both peritumoral and carcinoma tissues compared with healthy individuals, showing a correlation between tumor size and the likelihood of lymphatic metastasis [15]. However, the study did not specify the cellular source of mammaglobin expression in these tissues.

In contrast, in a study conducted by Leygue et al. using in situ hybridization on 13 breast tumor tissues, mammaglobin expression was found specifically in tumor epithelial cells, with no expression detected in stromal or inflammatory cells [57].

While both studies underscore the presence and potential relevance of mammaglobin in breast cancer tissues, they highlight different aspects: one emphasizes its potential prognostic value and correlation with certain tumor characteristics, while the other identifies the specific cell type expressing mammaglobin within the tumor. The apparent discrepancy in the cellular source might be due to differences in the methodologies used or the specific tissue samples analyzed in each study. 

As we delve deeper into the association between mammaglobin-A expression and various aspects of breast cancer, Table 2 provides a succinct summary of these relationships. The table categorizes key prognostic factors, including tumor subtype and stage, hormone receptor status, tumor grade, cell proliferation, and peritumoral expression. It outlines the correlation between these factors and mammaglobin-A expression, highlighting how varying expression levels can impact the pathogenesis and progression of the disease. This overview underscores the potential role of mammaglobin-A as a significant biomarker in breast cancer diagnosis and prognosis.

## 5. Mammaglobin Expression in Metastatic Breast Carcinoma: Potential Implications for Disease Detection and Monitoring

Breast cancer is a prevalent disease among women in the United States and is the second leading cause of cancer-related deaths [58]. While early-stage diagnoses provide hopeful prospects for successful treatment through surgical intervention, about half of the patients experience recurrence, often because of undetectable microscopic metastases present at the initial diagnosis stage [59,60]. These micrometastases, although clinically hidden, pose a significant challenge, as they can continue to grow and potentially lead to disease relapse, thus undermining the effectiveness of current treatment strategies [61].

To combat this, research aims to enhance the early detection and monitoring of micrometastases, which could allow for earlier intervention and potentially prevent disease recurrence [62]. Hence, advancements in diagnostic methods are crucial for identifying these metastatic cells and creating targeted therapeutic strategies [63]. Understanding the mechanisms underlying micrometastasis progression and developing novel detection and elimination strategies are vital areas of research, carrying the potential to significantly improve patient outcomes by reducing recurrence risk and offering more effective treatment options [64].

The metastasis process in breast tumors is complex and involves multiple biological events, all of which are influenced by various factors, critical for both the diagnosis and prognosis of the disease [65]. The overexpression of mammaglobin has been observed in metastatic breast carcinoma, suggesting its potential role in the metastatic process and indicating its potential use in monitoring disease progression and responses to treatment [7,66]. However, the mechanisms underlying mammaglobin’s overexpression in primary and metastatic breast carcinoma are still unclear, warranting further research to comprehend its role in the initiation, progression, and metastasis of breast carcinoma.

In an effort to understand the broad implications of mammaglobin in the metastatic process, we summarize the current knowledge about its role in different metastatic contexts in Table 3. This table underscores the value of mammaglobin as a marker, the methods of its detection, specific findings, and potential implications in the management of metastatic breast carcinoma.

In the following sections, we will further explore mammaglobin expression in different contexts, such as lymph node metastases, circulating tumor cells, and bone marrow metastases.

### 5.1. Mammaglobin: A Promising Marker for Lymph Node Metastasis in Breast Cancer

Metastasis detection in axillary lymph nodes is vital for breast cancer management, with a growing emphasis on sentinel lymph node biopsy for accurate prediction and clinical advantages [67]. Mammaglobin-A has emerged as a valuable marker for identifying breast cancer metastasis and micrometastases, offering superior sensitivity compared with other markers [7]. Its presence has been recognized in breast cancer-positive lymph node samples, whereas it is absent in normal or sentinel lymph nodes without detectable tumor cells [9]. The innovative use of a near-infrared fluorescent dye, VivoTag-S 680, conjugated to a monoclonal antibody against human mammaglobin-A, has shown promise in the non-invasive detection of metastasis in lymph nodes in animal models [68].

The application of multiple markers, including mammaglobin, cytokeratin 19, and carcinoembryonic antigen (CEA), has been proposed for guiding surgical decisions on axillary lymph node dissection in breast cancer [69,70]. In this context, while cytokeratin 19 has shown more specificity than mammaglobin in lymph node detection, mammaglobin’s detection in sentinel lymph nodes has been more informative than both cytokeratin 19 and CEA [71,72]. This suggests that mammaglobin could be a valuable molecular marker for detecting breast cancer in sentinel lymph nodes, potentially improving patient prognosis.

Innovative techniques like in vivo fluorescence imaging, leveraging a mammaglobin-A-specific monoclonal antibody conjugated to a near-infrared fluorescent dye, might reduce the need for surgical examination [73]. These techniques offer robust tools for studying tumor cells within the lymphatic system, detecting tumor cells in lymph nodes, and monitoring antitumor therapy [73]. Furthermore, the detection of minimal breast cancer mRNA markers like mammaglobin-A mRNA can aid in investigating node micrometastasis [12]. Real-time reverse transcription polymerase chain reaction (RT-PCR) can assist in detecting metastases in sentinel lymph nodes of breast cancer patients during surgery [12].

The use of RT-PCR to detect mammaglobin expression has been invaluable for identifying occult metastases in the lymph nodes of breast cancer patients [11]. Mammaglobin was first suggested as a marker for nodal metastases by Min et al., who found it to be expressed in all tested breast cancer cell lines but absent in normal lymph nodes [74]. Mammaglobin B (MAG-B) has also been identified as a useful marker for detecting histology-positive lymph node samples, despite its significant homology with mammaglobin-A [11].

Recent advancements in high-throughput RT-PCR protocols are promising for intraoperative sentinel lymph node analysis [70]. The practice of sentinel lymph node biopsy (SLNB) provides valuable prognostic information on metastatic spread with minimal associated morbidity [75]. In this regard, mammaglobin RT-PCR has displayed higher sensitivity compared with conventional histology [69]. Combining multiple markers, including mammaglobin, has been proposed for lymph node analysis, with real-time multigene RT-PCR assays showing excellent sensitivity and specificity [35]. Human mammaglobin (hMAM) mRNA overexpression has shown consistent alignment with lymph node metastasis, with hMAM protein levels found to be significantly high in patients with carcinoma in situ (CIS), invasive carcinoma (IC), and metastatic disease [49,51].

### 5.2. Circulating Mammaglobin: A Potential Diagnostic Marker for Breast Cancer

Mammaglobin has garnered growing research interest because of its presence in the bloodstream [12,15]. Notably, mammaglobin RT-PCR has been utilized to identify circulating mammary carcinoma cells in the blood samples of breast cancer patients [11,49]. One study reported sensitivity equivalent to detecting a single tumor cell among 10^6^–10^7^ white blood cells, with mammaglobin mRNA detected in 25% of patients, displaying variations based on clinical stages and disease status [76]. 

Researchers have also examined mammaglobin protein levels in the serum for potential diagnostic applications [11]. Utilizing monoclonal antibodies specific to recombinant mammaglobin, the protein was detected in 33% of primary breast cancer serum and 44% of metastatic breast cancer serum samples, suggesting that elevated mammaglobin protein levels could serve as a diagnostic marker [14]. 

#### Mammaglobin and Circulating Tumor Cells (CTCs): A Novel Diagnostic Avenue in Breast Cancer

Circulating tumor cells (CTCs), cancer cells that have detached from the primary tumor and entered the bloodstream, are significant contributors to cancer metastasis and are linked to worse clinical outcomes [77]. Recognized as crucial in cancer progression and patient prognosis, these cells provide insights into tumor molecular makeup and aid in personalized therapeutic decisions [78]. Mammaglobin, particularly its human form, hMAM mRNA, has gained attention as a potential marker for CTCs because of its breast-specific expression [49,53]. Elevated hMAM mRNA levels in leucocytes may indicate a higher number of CTCs, suggesting a higher risk of advanced disease progression [79].

The development of reverse transcriptase polymerase chain reaction (RT-PCR) technology has enabled the detection of specific tumor-related genes like mammaglobin in cancer patients’ blood, including those with metastatic breast cancer [53,80]. This has led to the identification and quantification of CTCs, offering valuable information about disease progression and overall survival [53]. However, hMAM mRNA detection methods can overlook tumor cells from hMAM-negative tumors, given that hMAM mRNA is only expressed in 70–80% of primary and metastatic breast cancer tissues [53]. To overcome this, a multi-gene RT-PCR assay has been proposed for the more sensitive and reliable detection of circulating breast cancer cells, showing increased detection sensitivity and providing significant prognostic information [53].

### 5.3. Mammaglobin: A Potential Indicator for Bone Marrow Metastasis in Breast Cancer

Breast cancer research advances have significantly improved patient survival rates over the past few decades, especially for localized disease [81]. However, survival rates decrease dramatically with metastatic lesions, highlighting the need for reliable early detection methods [82]. Mammaglobin, a breast-tissue-specific antigen found in the bone marrow of breast cancer patients, is under investigation as a potential marker for breast cancer metastasis given its higher expression rates in patients with metastasis and progressive disease [83].

While some reports suggest that cytokines might induce mammaglobin expression in non-cancerous bone marrow and peripheral stem cells, leading to false positives, research by Silva et al. did not detect mammaglobin transcripts in bone marrow samples from normal donors [83,84]. Instead, they identified mammaglobin expression in bone marrow aspirates from breast cancer patients, particularly those with metastatic disease [84]. Despite mammaglobin’s potential as a metastasis marker, more research is needed to confirm its specificity and establish its clinical utility, especially as immunohistochemical staining for MAM-A in bone marrow could be more sensitive in detecting early bone marrow micrometastases.

## 6. Mammaglobin in Breast Cancer: A Multifaceted Entity

In the complex and multifarious landscape of breast cancer, mammaglobin plays a critical yet not entirely understood role [9]. This member of the secretoglobin family has shown significant promise in various capacities, from contributing to our understanding of the biological underpinnings of breast cancer to potentially guiding prognosis and therapy [51]. In the following sections, we delve into the intricate biological function of mammaglobin and how its varied expression might impact the prognosis of breast cancer patients. We will further explore mammaglobin’s potential role as a therapeutic target, while also examining its expression in other carcinomas. By comprehensively examining these different dimensions, we aim to offer a nuanced perspective of mammaglobin in the context of breast cancer.

### 6.1. The Biological Function of Mammaglobin in Breast Carcinoma: A Dual Role

The role of mammaglobin, especially mammaglobin-A, in breast cancer is complex, modulating processes like cell proliferation, migration, and invasion [8]. Although it exhibits both pro-tumor and anti-tumor activities, the mechanisms behind the impact of its upregulation on cancer aggressiveness remain to be fully understood (Figure 1). A study by Picot et al. suggested that mammaglobin-A can promote tumor cell proliferation, migration, and invasion, activating several critical signaling pathways such as mitogen-activated protein kinase (MAPK), focal adhesion kinase (FAK), matrix metalloproteinases (MMPs), and nuclear factor kappa B (NF-κB), all essential in cancer development and metastasis [85].

Mammaglobin-A also seems to regulate the epithelial-to-mesenchymal transition (EMT), a process tied to increased cancer cell invasiveness and metastatic potential [17]. This idea is supported by the observed suppression of mesenchymal-related genes like Zinc Finger E-box-binding Homeobox 1 (ZEB1), Snail, and Twist following the inhibition of mammaglobin-A expression. Mammaglobin-A (MAG-A) also appears to play a significant role in breast cancer cell growth and survival, with the suppression of MAG-A expression in the aggressive MB231 breast cancer cell line resulting in attenuated cell growth [85].

Additionally, reduced MAG-A expression has been linked to decreased cell migration, invasion capacity, and anoikis resistance in the MB231 cell line [85]. This reduction correlates with decreased matrix metalloproteinases (MMP-9 and MMP-2) and decreased focal adhesion kinase (FAK) activity, proteins known for promoting invasive behavior [85]. MAG-A also seems to influence gene expression profiles associated with epithelial-mesenchymal transition (EMT) and mesenchymal–epithelial transition (MET) [85].

Interestingly, decreased mammaglobin expression has been associated with increased resistance to apoptosis and altered responses to chemotherapeutic drugs in breast cancer cells [85]. Furthermore, chronic treatment with trastuzumab, a common breast cancer drug, has been linked to increased MAG-A expression in HER2-positive/ER-negative breast cancer cells, potentially enhancing their proliferation, migration, and invasion capacities and potentially contributing to trastuzumab resistance [86]. In conclusion, mammaglobin, particularly MAG-A, plays a multifaceted role in breast cancer, influencing tumor aggressiveness, chemosensitivity, apoptosis resistance, and metastatic disease progression, which warrants further investigation.

### 6.2. Expression of Mammaglobin-A in Other Carcinomas

Mammaglobin-A, a protein markedly present in an array of tumors, is predominantly observed in cancers originating from the salivary glands, breasts, endometrium, and ovaries [8]. Its detection relies on a method known as immunohistochemistry. Nearly 99% of tumors that test positive for mammaglobin-A are found within these four organs [8]. Nonetheless, this protein is also detected, although to a lesser degree, in other, less frequent cancer types [8].

Aside from breast cancer, varying degrees of mammaglobin-A presence have been documented in cancers like uterine endometrium carcinomas, ovarian carcinomas, and prostatic adenocarcinomas [8]. The observed variations in mammaglobin-A expression are attributed to differences in the antibodies used, the immunostaining protocols implemented, and the criteria set for defining the staining outcomes. Remarkably, there is an absolute absence of mammaglobin-A immunostaining in several non-breast and non-gynecological cancers, underlining the diagnostic utility of mammaglobin-A immunohistochemistry, especially for gynecological tumors [8].

Interestingly, studies have shown links between low or non-existent mammaglobin-A immunostaining and advanced tumor stages across various types of cancers [12]. However, these associations do not necessarily translate to correlations with overall survival. Differing results across various studies examining mammaglobin-A expression in tumors are primarily due to disparities in the antibodies used, staining protocols followed, and what constitutes a “positive” case. Despite this, the expression pattern of mammaglobin-A in tumors generally mirrors that in normal tissues, providing invaluable insights into the role of mammaglobin-A across a spectrum of cancers [8].

### 6.3. Mammaglobin-A as a Promising Therapeutic Target in Breast Cancer

Mammaglobin-A (MAM-A), a secretoglobin protein consisting of 93 amino acids, offers promising potential as a target for breast cancer vaccine therapy because of its high expression in breast cancers and minimal presence in healthy tissues [87]. Its high immunogenicity allows for the generation of MAM-A-specific CD8^+^ and CD4^+^ T cells that recognize and target MAM-A-expressing breast cancers [88]. However, the endogenous immune response to MAM-A seems insufficient to eliminate developing breast cancers, potentially because of insufficient MAM-A-specific T cells, inadequate infiltration into the tumor microenvironment (TME), or downregulation due to immunoregulatory elements [87].

Previous research has managed to induce MAM-A-specific CD8^+^ T cells through DNA vaccination, with a phase 1 clinical trial involving 15 patients demonstrating no severe toxicities and an increase in peripheral MAM-A-specific T cells [89]. This trial also reported an increase in ICOS^hi^CD4^+^ T cells and a decrease in Foxp3^+^CD4^+^ T cells six months post-vaccination [89]. In eight vaccinated patients expressing HLA-A 0201, a significant increase in MAM-A-specific CD8 T cells was noted, and their cytotoxic activity against breast cancer cell lines displayed efficacy dependent on certain parameters [89].

The effectiveness of MAM-A-specific CD8^+^ T cells hinges not only on population expansion but also on the modulation of regulatory networks within the TME [87]. Associations have been made between lower levels of certain tumor-associated macrophages and lymphocytes and longer recurrence-free survival [87]. Despite encouraging preliminary results, further research is needed to elucidate interactions between the MAM-A vaccine and the TME’s regulatory networks and to explore the specific signaling pathways involved in the expression of MAM-A in breast cancer tissues.

### 6.4. Mammaglobin as a Potential Target for Breast Cancer Immunotherapy

Mammaglobin, a mammary-specific antigen, has been spotlighted as a promising target for breast cancer immunotherapy because of its elevated expression in breast cancer tissues relative to normal tissues [9]. This heightened expression enhances the prospects of developing effective targeted therapies, including vaccine-based approaches such as the MAM-A DNA vaccine [90,91]. Such vaccines are crafted to stimulate the immune system, prompting it to recognize and attack tumor cells by presenting tumor-associated antigens. The overarching goal is to halt tumor progression and eradicate existing malignancies, offering a new horizon in the therapeutic landscape for breast cancer.

Human MAM-A holds promise in spearheading effective therapies and vaccines against breast cancer cells. This potential arises from targeting mammaglobin-derived epitopes on cytotoxic T-lymphocytes for the effective delivery of targeted interventions. The strategic focus includes utilizing a transmembrane N-terminal domain of MAM-A or MAM-A-derived epitopes in association with HLA-A-2, HLA-A-3, and HLA-A-24 to evoke CD8-restricted responses [90,91,92,93,94,95].

Being a membrane-associated protein, MAM-A might serve as an invaluable molecular marker for devising targeted drug therapies specific to breast cancer [96]. Furthermore, its potential is also underscored in crafting experimental vaccines against breast cancer cells [90]. Techniques such as stimulating CD4^+^CD25^−^ T cells in vitro with MAM-A-pulsed antigen-presenting cells or transducing dendritic cells with Tat-MAM-A have been explored [91,97]. Notably, the ability of MAM-A to bind to breast cancer cells has also raised intriguing propositions regarding its potential role in guiding radioisotopes or toxins to specifically target these malignant cells [98].

Several therapeutic strategies leveraging MAM-A have been proposed, including the innovative use of the MAM-A promoter to facilitate gene therapy. This approach might deliver oncolytic viruses or toxic genes directly to mammary tumors [99]. Additionally, combining targets like MAM-A with other markers such as HER-2/neu in active immunotherapy could potentially amplify the therapeutic vaccine’s efficacy and bolster specific T cells for adoptive immunotherapy, especially in treating established metastatic diseases [100]. It is worth noting, however, that recent findings suggest inducing the overexpression of MAM-a in breast cancer cells might diminish their metastatic potential, offering another intriguing avenue for managing aggressive forms of breast cancer [101].

## 7. Current Challenges and Future Perspectives

Despite the optimistic potential of mammaglobin as a breast carcinoma biomarker, several crucial challenges persist:Sensitivity and Specificity: Enhancing the sensitivity and specificity of mammaglobin detection in breast carcinoma is imperative. Future studies need to concentrate on refining the techniques used in mammaglobin detection to boost accuracy.Method Standardization: Currently, a universal protocol for detecting and quantifying mammaglobin across diverse sample types is absent, creating hurdles for comparison between studies. The establishment of shared protocols should be prioritized.Biological Role Comprehension: The biological role mammaglobin plays in breast carcinoma is not entirely clear. Clarifying its exact function could offer critical insights into its effectiveness as a diagnostic and prognostic marker and potential as a therapeutic target.

Moving forward, the realm of mammaglobin research reveals a number of thrilling prospects:Early Detection and Diagnosis: Enhanced detection methods could pave the way for the use of mammaglobin as a non-invasive biomarker for the early detection and diagnosis of breast carcinoma.Prognostic Marker: Mammaglobin may become a valuable prognostic marker if larger prospective studies substantiate the link between its levels and disease progression/outcomes.Personalized Therapy: If mammaglobin is proven to be integral to the pathogenesis of breast carcinoma, it could be considered a target for personalized therapies.Companion Diagnostic Tool: In conjunction with other established biomarkers, mammaglobin could serve as a companion diagnostic tool to improve diagnostic accuracy and inform treatment choices.

The potential for mammaglobin to serve as a biomarker for breast carcinoma holds promise, yet its realization hinges on rigorous, continued research to surmount current obstacles and validate its clinical application. Should these challenges be resolved, mammaglobin could herald a new phase in breast carcinoma management, offering tools for early detection, precise prognosis, and personalized treatment.

Table 4 summarizes the main challenges and prospects associated with mammaglobin as a biomarker for breast carcinoma. It provides an overview of current obstacles to its implementation, as well as its potential future applications, highlighting the need for ongoing research in this field.

## 8. Conclusions

Mammaglobin, a breast-tissue-specific biomarker, offers transformative potential in breast carcinoma diagnosis, prognosis, and treatment, albeit with a few challenges. While its detection in various samples, including tumor tissues, peritumoral regions, and circulating blood, promises a broad application, improving its detection sensitivity and specificity, standardizing detection protocols, and understanding its biological role are essential steps toward realizing its full potential. Despite these hurdles, mammaglobin research holds exciting promise for breast carcinoma management, offering a vision of early detection, accurate prognosis, and personalized treatment. In the face of challenges and opportunities, continued investment in mammaglobin research is essential, driving us toward the ultimate goal of improved patient outcomes in breast carcinoma.

## Figures and Tables

**Figure 1 ijms-24-13407-f001:**
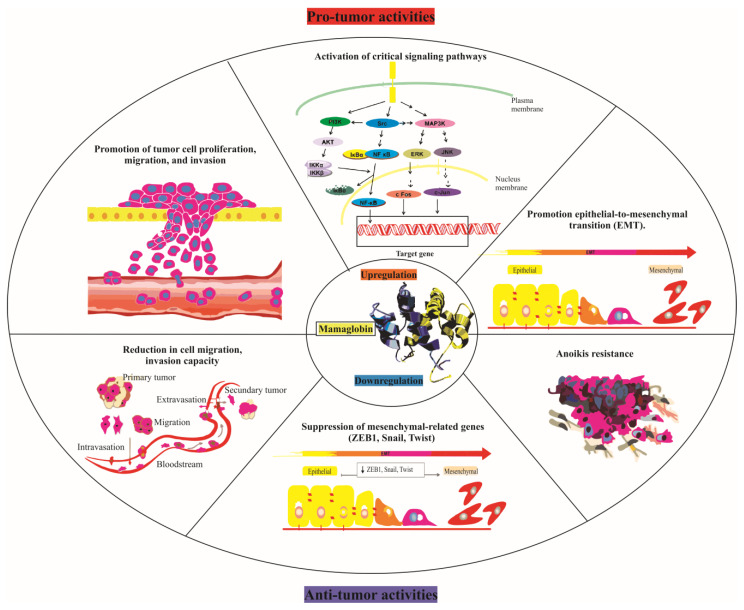
Pro-tumor and anti-tumor activities of mammaglobin-A. The upper half of the figure depicts the pro-tumor activities triggered by mammaglobin-A upregulation. This includes the promotion of tumor cell proliferation, migration, and invasion; the activation of critical signaling pathways such as MAPK, FAK, MMPs, and NF-κB; and the initiation of epithelial-to-mesenchymal transition (EMT). Conversely, the lower half illustrates the anti-tumor activities resulting from mammaglobin-A downregulation. This encompasses the suppression of mesenchymal-related genes (ZEB1, Snail, Twist) and a decrease in cell migration, invasion capacity, and anoikis resistance subsequent to hMAG-A suppression. This figure thereby underlines the dual and contrasting roles of mammaglobin-A in tumor dynamics, dependent on its regulation.

**Table 1 ijms-24-13407-t001:** Comparative analysis of human mammaglobin and other breast cancer markers.

Tumour Marker	Specific Remarks
Carcinoembryonic antigen (CEA)	Noted for its involvement in recurrence and correlation with circulating tumor cell discovery, yet its sensitivity is considered low [23].
Cytokeratins (CK19 and CK20)	Known for their low sensitivity, these markers can be found in both normal cells and a variety of tumors [30].
Epidermal growth factor receptor (EGFR)	Similar to cytokeratins, EGFR presents with low sensitivity and can be identified in normal cells and various tumors [31].
Maspin	This marker is associated with a reduced risk of recurrence [32].
Polymorphic epithelial mucin (MUC-1)	Adverse outcomes are linked to high pre-operative CA 15-3 levels. CA27-29 offers little utility. Low sensitivity and expression in normal cells and hematological tumors are its key characteristics [33,34].
B726P	When used in tandem with hMAG, B726P could aid in distinguishing between mammary and non-mammary tissues [35].
Urokinase plasminogen activator (uPA)	The presence of this marker might provide valuable information for prognosis [36].
Plasminogen activator inhibitor 1 (PAI-1)	Similar to uPA, PAI-1 can be helpful in determining prognosis [37].
Estrogen receptor (ER)	Detectable in primary lung adenocarcinomas, ER is used for predicting hormonal therapy responses in breast cancer despite its limited prognostic significance [38].
Progesterone receptor (PR)	PR is considered a key factor for hormonal therapy [39].
Human epidermal growth factor receptor-2 (HER-2)	Human epidermal growth factor receptor-2 (HER-2) is highly expressed in breast cancers with an amplified ERBB2 gene, i.e., those of the HER2 molecular subtype, making HER-2 instrumental in the selection process for Herceptin therapy [40].
Breast cancer 1 and 2 early onset (BRAC-1 and BRAC-2)	These markers can assist in identifying high-risk patients [41].
Small breast epithelial mucin (SBEM)	SBEM is detectable in roughly 52% of breast tumors, with no presence in non-breast tumors [42].
Survivin	This marker does not have specificity for breast cancer [43].
Ki67	Ki67 is thought to act as an indicator of breast cancer progression [44].
Gross cystic disease fluid protein 15 (GCDFP-15)	This marker is noted for its significant link with mammary differentiation and has shown a correlation with mammaglobin expression. Research is ongoing into its potential as a breast cancer biomarker [45].
Human mammaglobin (hMAG)	hMAG exhibits high expression (80–90%) in breast tumors and is particularly sensitive (97%) in detecting residual disease [9].

**Table 2 ijms-24-13407-t002:** Mammaglobin-A expression and correlation with breast cancer prognostic factors.

Prognostic Factor	Mammaglobin-A Expression
Tumor Subtype and Stage	Variable expression with positivity rates ranging from 59% to 100% for lobular breast carcinomas and 25% to 94% for invasive breast carcinomas
Hormone Receptor Status	Elevated levels correlate with estrogen receptor (ER) and progesterone receptor (PR) status
Tumor Grade	High tumor grade associated with overexpression of human mammaglobin (hMAM)
Cell Proliferation	Higher Ki-67 proliferation index observed in hMAM-positive invasive breast cancer
Peritumoral Expression	Presence detected, suggesting a possible link with local invasion, metastasis, and aggressive disease phenotypes

**Table 3 ijms-24-13407-t003:** Mammaglobin as a marker in different contexts of metastatic breast carcinoma.

Metastatic Context	Method of Detection	Specific Findings	Potential Implications
Lymph Node Metastasis	VivoTag-S 680, RT-PCR	Presence in positive lymph nodes, absent in normal/sentinel nodes	Guiding surgical decisions, improved prognosis
Circulating Mammaglobin	RT-PCR	Detected in 25% of patients	Potential diagnostic marker
Circulating Tumor Cells (CTCs)	RT-PCR (hMAM mRNA)	hMAM mRNA expressed in 70–80% of breast cancers	Indicates a higher number of CTCs and a higher risk of advanced disease progression
Bone Marrow Metastasis	Bone marrow aspirates, immunohistochemical staining	Mammaglobin expression is higher in patients with metastasis	Potential marker for metastasis

**Table 4 ijms-24-13407-t004:** Challenges and prospects of mammaglobin in breast carcinoma.

Area	Challenge/Prospect	Description
Sensitivity and Specificity	Challenge	Enhancing the detection accuracy of mammaglobin in breast carcinoma
Method Standardization	Challenge	Establishing a universal protocol for detecting and quantifying mammaglobin
Biological Role Comprehension	Challenge	Understanding the exact role of mammaglobin in breast carcinoma
Early Detection and Diagnosis	Prospect	Using mammaglobin as a non-invasive biomarker for early detection and diagnosis
Prognostic Marker	Prospect	Establishing mammaglobin as a valuable prognostic marker through larger prospective studies
Personalized Therapy	Prospect	Exploring mammaglobin as a potential target for personalized therapies
Companion Diagnostic Tool	Prospect	Utilizing mammaglobin alongside other established biomarkers to improve diagnostic accuracy and inform treatment choices

## Data Availability

As this is a review article, it does not contain any new, original research data. Instead, it synthesizes and analyzes previously published data and information. All data and materials discussed in this review are cited appropriately and can be found in the referenced articles and resources. Therefore, a data availability statement is not applicable to this paper.

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
