# Peer review of "The Enigma of Mammaglobin: Redefining the Biomarker Paradigm in Breast Carcinoma"

_ijms, 2023, doi:10.3390/ijms241713407_

Round 1
Reviewer 1 Report
1/ The title Mammaglobin: An Overview seems an overlap with the introduction part and this section can be squeezed into introduction. The expression pattern can stand alone as a subtitle.
2/ The application of mammaglobin in clinical setting is worthy to discuss and many works have been done on its role in immunotherapy.
Author Response
Response to Reviewer's Comments:
Comment 1: The title "Mammaglobin: An Overview" seems an overlap with the introduction part and this section can be squeezed into the introduction. The expression pattern can stand alone as a subtitle.
Response: Thank you for this suggestion. We have revised the manuscript accordingly and integrated the content of "Mammaglobin: An Overview" into the introduction. Additionally, we have established "Expression Pattern" as a standalone subtitle, as you recommended.
Comment 2: The application of mammaglobin in the clinical setting is worthy to discuss and many works have been done on its role in immunotherapy.
Response: We appreciate your insight. We have now dedicated a specific section in our manuscript to thoroughly discuss the role of mammaglobin-A in the immunotherapy of breast cancer. This section delves into the recent advancements and contributions of mammaglobin-A in this therapeutic realm.
We would like to express our sincere gratitude to the reviewer for their constructive feedback and invaluable insights, which significantly enhanced the quality and coherence of our manuscript. Your expertise and thoughtful suggestions have been instrumental in refining our work, and we deeply appreciate the time and effort invested in reviewing our manuscript.
Reviewer 2 Report
General comments
- The study titled "The Enigma of Mammaglobin: Redefining the Biomarker Paradigm in Breast Carcinoma" plays an important role in looking for more breast cancer biomarkers that could be used as possible therapy targets.
- The manuscript's overall structure is acceptable.
Minor comments
- As a review paper, suggest using references that are the most recent and haven't been published for longer than 5 to 10 years.
- In the review to make it attractive, if including any clinical trial data that shows mammaglobin targeting, including chemotherapy and or vaccination, would be better.
Author Response
Response to Reviewer's Comments:
Comment: As a review paper, suggest using references that are the most recent and haven't been published for longer than 5 to 10 years.
Response: In response to the reviewer's comment, we made a concerted effort to use references that are within the 5 to 10-year range. However, there are certain seminal works and foundational studies that are older than 10 years which we believed were essential to provide a comprehensive understanding of the topic. We felt these references were indispensable for contextualizing our review and ensuring a holistic coverage of the subject matter.
Comment: In the review to make it attractive, if including any clinical trial data that shows mammaglobin targeting, including chemotherapy and or vaccination, would be better.
Response: We value this suggestion. Accordingly, we have dedicated a specific section in our manuscript to delve into the application of drugs and vaccines targeting mammaglobin. This section emphasizes the recent advancements in clinical trials, showcasing the potential and efficacy of mammaglobin-targeted therapies.
We deeply appreciate the reviewer's meticulous feedback and constructive suggestions, which have been crucial in enhancing the depth and quality of our manuscript. The time and expertise devoted to reviewing our work are highly valued, and we are thankful for your contributions towards refining our paper.